

# Adherence to the Mediterranean diet, kinanthropometric characteristics and physical performance of young male handball players

David Romero-García[1], Francisco Esparza-Ros[2], María Picó García[1], José Miguel Martínez-Sanz[1] and Raquel Vaquero-Cristóbal[3]

[1] Nursing Department, Food and Nutrition Research Group (ALINUT), Faculty of Health Sciences, University de Alicante, San Vicente del Raspeig, Alicante, Spain
[2] Research Group on Prevention of Sports Injuries, International Chair of Kinanthropometry, Catholic University of Murcia, Murcia, Spain
[3] Research Group on Prevention of Sports Injuries, Faculty of Sport, Catholic University of Murcia, Murcia, Spain

Corresponding author
José Miguel Martínez-Sanz,
josemiguel.ms@ua.es

## ABSTRACT

**Introduction.** Handball is a team sport whose performance depends on a large number of factors. The objectives of the present study were to analyze the differences in physical performance, kinanthropometric variables, and adherence to the Mediterranean diet (MD), in handball players according to sports category, as well as the possible relationships between these variables.

**Methods.** One hundred and thirty-three male handball players (35 infant; 46 cadets; 26 juniors and 26 seniors players) underwent a kinanthropometric analysis following the ISAK protocol, self-completed the KIDMED questionnaire, and their physical condition was evaluated with different physical tests.

**Results.** Significant differences ($p < 0.001$–$0.007$) were found between the sports categories in most of the physical condition variables and anthropometric characteristics, but not in the degree of adherence to the MD. The predominant level of adherence to the MD was medium (47.4%), followed by good (42.1%), and correlated with the body mass, the height, the BMI, the muscle mass and the $\sum$3-girths sum corrected, but not with physical test results. A multiple linear regression analysis showed that the fat mass and muscle mass variables had a more specific weight in the results of the physical fitness tests.

**Conclusions.** There are differences according to sports category in kinanthropometric characteristics and physical fitness tests in adolescent handball players. The predominant degrees of adherence to the MD were medium and good. A relationship was found between anthropometric characteristics and physical performance in handball players.

# INTRODUCTION

Handball is a complex, multifactorial, and high-contact team sport, in which different actions, movements or plays occur intermittently at very high intensity (*Martínez-Rodríguez*

*et al., 2020*). These periods alternate with brief moments of low intensity that serve as a recovery period before the next play (*Hermassi, Laudner & Schwesig, 2019*; *Hermassi et al., 2020*). All these situations are influenced by the physical capabilities of the players themselves highlighting strength, endurance, speed, and flexibility (*Hammami et al., 2019*; *Molina-López et al., 2020*).

Considering the characteristics of the game, there is no doubt that handball players need to have a correct physical preparation to be able to overcome all the physical demands and supply the energy required during a training session or match. Such physical preparation is related to the anthropometric characteristics and body composition of the individual players, which influence their performance during competition (*Ghobadi et al., 2013*; *Martínez-Rodríguez et al., 2020*). In fact, both the anthropometry and body composition of the players are important factors that determine performance within a team (*Matthys et al., 2013*).

Generally speaking, players that make up elite teams are tall, with low body fat values, and good muscle development (*Vila et al., 2012*; *Ghobadi et al., 2013*). In fact, greater muscle mass also implies improved performance, probably due to increased maximal strength and muscle power (*Granados et al., 2015*). However, researchers have not yet been able to establish a consensus to be able to exactly define which anthropometric characteristics are more related or have a greater influence on the improvement of performance in handball players.

Another factor that may have a strong influence is the athlete's diet (*Kerksick et al., 2018*). In fact, organizations with great international prestige insist that the performance of athletes, as well as their recovery, can be improved with optimal nutrition (*Thomas, Erdman & Burke, 2016*). With respect to nutrition, it is known that the Mediterranean diet (MD) is a healthy diet model which is characterized by a high consumption of fruits, vegetables, cereals, nuts, legumes and olive oil; a moderately high consumption of fish; a moderate consumption of dairy products; and a low consumption of meat, poultry and foods with saturated fats. A good adherence to MD has been associated with certain physical benefits and high levels of health-related quality of life (*Galan-Lopez et al., 2018*; *Galan-Lopez et al., 2019*; *Galan-Lopez et al., 2020*; *Manzano-Carrasco et al., 2020a*). This could be because good adherence to this dietary pattern provides a balanced intake of macronutrients such as complex carbohydrates, unsaturated fats and high biological value proteins, as well as micronutrients such as vitamins and minerals (*Davis et al., 2015*). This adequate intake of macronutrients and micronutrients could favor the energy availability of athletes during competition, promote physiological processes and prevent a positive energy balance that leads to fat accumulation (*Tosti, Bertozzi & Fontana, 2018*). However, very few studies have analyzed adherence to the MD in young athletes, and how it influences sports performance, body composition characteristics, or both, not finding a high adherence of young athletes to this dietary pattern. (*Rubio-Árias et al., 2015*; *Vaquero-Cristóbal et al., 2018*; *Manzano-Carrasco et al., 2020b*; *Martínez-Rodríguez et al., 2021*). In addition, no studies have been found about handball players.

Therefore, the objectives of the present study were: (a) to observe whether there is any difference between anthropometric characteristics, sports performance, and adherence

to the MD depending on the sports category of handball players; (b) to know the degree of adherence to the MD of the players; (c) to know whether this degree of adherence to the MD influences sports performance and anthropometric characteristics; (d) to know whether there is a relationship between anthropometric characteristics and sports performance. The hypotheses of this study were: (1) athletes of higher categories would show a greater optimization of anthropometric characteristics and better results in the physical condition tests than their equivalent athletes of lower categories; (2) adolescent handball players would have a heterogeneous adherence to DM, with no clear pattern in this regard; (3) a better adherence to DM would positively influence the presence of more optimal anthropometric parameters and sports performance; and (4) adolescent handball players with better body composition, that is, with greater muscle development and a lower adipose component, and with good adherence to DM will have better results in physical fitness tests related to handball performance.

## MATERIALS & METHODS

### Study design

The present study was carried out by means of a descriptive-correlational cross-sectional study in which anthropometric characteristics, adherence to the Mediterranean diet, and performance in different physical tests, were evaluated in young handball players. The sample size calculation was performed with Rstudio software (version 3.15.0; *RStudio Team, 2018*). The significance level was set a priori at $\alpha = 0.05$. The standard deviation (SD) was set according to the muscle mass data from previous studies in adolescent athletes (SD = 2.59) (*Albaladejo-Saura et al., 2022*). With an estimated error (d) of 0.44 kg, the sample size needed was 133 subjects. The study population was selected through non-probabilistic convenience sampling among clubs with infant, cadet, junior and senior categories in the provinces of Alicante and Murcia (Spain).

### Participants

One hundred and thirty-three male handball players between 12 and 28 years of age voluntarily participated in this study. They were divided according to the category in which they competed, with the final sample being 35 infant (mean age: 13.41 ± 0.40 years), 46 cadets (mean age: 14.83 ± 0.64 years), 26 juniors (mean age: 17.20 ± 0.55 years), and 26 seniors (mean age: 20.09 ± 2.76) years. The inclusion criteria for the study were: (a) at least two years playing handball; (b) being federated in handball; (c) training a minimum of three days per week; and (d) training for at least one month without missing any training. The exclusion criteria for the study were: (a) being injured at the time of the evaluations.

### Procedure

The study was approved by the Ethics Committee of the University of Alicante (code: UA-2022-02-01). In addition, all the players were previously informed about the objective and method of the study, and signed the informed consent before starting the study, or their parents or legal guardians, failing this, in the case of minors.

The measurements were performed in the same week in three evaluation sessions, leaving one rest day between the second and third evaluation day. On the measurement

days, the players should not have performed high-intensity exercise the previous day, nor should they have performed training or stretching sessions on the same day. On the first day, kinanthropometric characteristics and adherence to the MD were evaluated; on the second day, agility and endurance tests were performed; and on the last day, jumping, throwing, and sprinting tests were completed, always leaving five minutes of rest between tests.

### Adherence to the Mediterranean diet

To analyze adherence to the MD, participants self-completed the questionnaire that assessed MD quality index in children and adolescents (KIDMED, KM). With a maximum score of 12 points, responses were categorized according to the level of quality of MD: ≤3 poor quality, 4–7 medium quality and ≥8 optimal quality (*Serra-Majem et al., 2004*).

### Kinanthropometric evaluation

The kinanthropometric evaluation was performed by an anthropometrist level 2 accredited by the International Society for the Advancement of Kinanthropometry (ISAK), with a technical measurement error of 0.03% for basic measurements, 2.24% for skinfolds and 0.36% for girths, and following the international standards recommended by the ISAK (*Esparza-Ros, Vaquero-Cristóbal & Marfell-Jones, 2019*). The following variables were taken: the four basic measurements (body mass, height, sitting height y arms span), three skinfolds (triceps, thigh and leg), and four girths (relaxed and contracted arm, thigh and leg). For the determination of body mass, a SECA 862 scale (SECA, Hamburg, Germany), with an accuracy of 100 g was used; for height and sitting height, a SECA 217 (SECA, Hamburg, Germany) measuring rod with a one mm accuracy was used; and for wingspan, an Avanutri wingspan meter (Avanutri, Tres Rios, Brazil); for girths, a CESCORF metal tape (CESCORF, Portro Alegre, Brazil) was used; and for skinfolds, a Slimguide plicometer (Creative Health Products, United States) with a 0.5 mm precision was utilized. All the anthropometric measurements were taken two or three times, depending on whether the difference between the first two was greater than 5% for skinfolds, and 1% in the rest of the measurements, taking the mean or median, respectively, for subsequent analyses. The temperature of the room where the measurements were taken was standardized at 24 °C, and all measurements were taken from 15:00 to 21:00.

With the data collected, the following were estimated: body mass index (BMI), with the formula body mass (kg)/height (m)2 (*Alvero Cruz et al., 2010*); the percentage of fat mass (FM) (*Slaughter et al., 1988*) and muscle mass (MM) (*Poortmans et al., 2005*), with the formulas chosen because they have been validated in a growing population (*Alvero Cruz et al., 2010*); $\sum$3 skinfolds (triceps, thigh and leg); and $\sum$3-girths sum corrected (relaxed arm, thigh and leg), with the formula corrected perimeter = perimeter - $\pi$ × fold (*Alvero Cruz et al., 2010*).

### Physical condition

The T-Half Test protocol (*Sassi et al., 2009*) was used to evaluate the agility of the players. This test was chosen because unlike the T-Test, it is shorter in distance and

more representative for handball. To monitor the start and end of the test, a photocell (Witty, Microgate, Italy) was used.

The endurance capacity of the players was evaluated following the protocol of the Intermittent Recovery Yo-Yo level 1 test (*Krustrup et al., 2003*). In addition to recording the total meters run by each athlete in this test, the theoretical $VO_2$ max was estimated using the equation of *Bangsbo, Iaia & Krustrup (2008)*, which has been previously used in similar contexts (*Gonçalves & Carvalho, 2021*; *Macpherson & Weston, 2015*; *Köklü et al., 2012*).

To assess the jumping ability of the players, they were asked to perform the Squad Jump (SJ) and Counter Movement Jump (CMJ). The latest version of the MyJump application (*Balsalobre-Fernández, Glaister & Lockey, 2015*) was used to measure the jumps, and a previously described protocol was used (*Hermassi et al., 2021*).

To assess the players' upper limb power, they were given the overhead medicine ball throw test, and a tape measure (Haest, Germany) with 0.1 cm accuracy was used to measure the distance (*Albaladejo-Saura et al., 2022*).

For the assessment of speed, a 30-m sprint was performed at the maximum possible intensity. To monitor the start and end of the test, two photocells (Witty, Microgate, Italy) were used.

The players performed two valid attempts of the agility, jump, throw, and sprint tests, and one attempt of the endurance test. A familiarization with the physical test protocols was performed the week before the measurements. In addition, before sessions 2 and 3, a standardized warm-up was performed consisting of 10 min of continuous running at a gentle jog, dynamic joint mobility exercises and low-intensity exercises, simulating the tests they were going to perform in that session. The researchers encouraged the players to perform the physical tests at the maximum possible intensity.

## Statistical analysis

Both the kinanthropometric characteristics of the players, as well as adherence to the MD and the results obtained in the performance tests, were analyzed using descriptive statistics, to obtain the mean and standard deviation. The Kolmogorov–Smirnov normality test was performed to evaluate that all variables had a normal distribution. The kurtosis was also evaluated, and the Mauchly sphericity test was performed to test the sphericity of the data. Given the normal distribution of the data and to analyze the differences between the categories in the anthropometric variables, adherence to MD, and the results of the physical tests, an analysis of variance (ANOVA) was performed, as well as an analysis of covariance (ANCOVA) to observe the influence of the covariate age. When significant differences were found between groups, a pairwise comparison was performed using the Bonferroni correction for multiple comparisons. In addition, a bivariate correlation was performed between the kinanthropometric characteristics of the players and adherence to MD with the results obtained from the physical tests. The association between these variables was measured as follows: $r < 0.3$, low association; $r = 0.3$–$0.5$, moderate association; and $r > 0.5$, high association. Finally, a stepwise linear regression analysis was performed on all the variables that were found to correlate with the physical fitness tests. As for the

categorical variables, a chi-square test ($\chi 2$) was performed to analyze the distribution of the degree of adherence to the Mediterranean diet. The minimum level of statistical significance was set at $p < 0.05$. All the data were analyzed using the Statistical Package for the Social Sciences (SPSS) version 25.0 (IBM, Armonk, NY, USA).

## RESULTS

Table 1 shows the descriptive statistics (mean ± standard deviation) of all measured variables, as well as the differences between groups, the main effects of the covariate age and the interaction of the category*age model. When comparing results among categories, significant differences ($F = 4.259-58.408$; $p < 0.001-0.007$) were found for all measured variables, with the exception of the KM score ($F = 1.773$; $p = 0.156$). When comparing participants according to age, significant differences were found in all variables ($F = 78.554-9.563$; $p < 0.001-0.002$), except for FM, $\sum$3-skinfolds, KM score and the distance and the VO2max in the YoYo test ($F = 3.603-0.266$; $p = 0.060-0.607$). The interaction between category and age also showed significant differences in all measured variables ($F = 43.773-3.196$; $p < 0.001-0.015$), except for the KM score ($F = 1.446$; $p = 0.223$).

With respect to the anthropometric variables (Table 2), it was found that the infant category showed significantly lower values with the majority of the other categories in body mass, height, BMI, MM and girths corrected sum, and significantly higher values respect the majority of the other categories in the percentage of FM and $\sum$3-skinfolds ($p < 0.002-0.021$. The cadet category compared to the junior category, showed significantly lower values in body mass, height, MM and $\sum$3-girths sum corrected ($p < 0.001-0.009$). Regarding the comparison between cadets and seniors, it was found that cadets showed significantly lower values in body mass, BMI, MM and $\sum$3-girths sum corrected ($p < 0.001$). No significant differences were found between junior and senior categories ($p > 0.05$).

Regarding the differences in the physical fitness variables between categories (Table 3), the infant category showed a worse sports performance in the medicine ball throw, T-Half test, SJ, CMJ, sprint and Yo-Yo test variables ($p < 0.000-0.026$). The cadet category compared to the junior category, showed significantly worse performance in the medicine ball throw, T-Half test, SJ y CMJ ($p < 0.001-0.005$). Regarding the comparison between cadets and seniors, it was found that cadets showed significantly lower distance in the medicine ball throw ($p < 0.001$). On the other hand, in the comparison of the junior and senior categories, junior category showed significantly higher differences in the SJ jump test ($p < 0.003$). The category*age group interaction showed that age had a significant influence ($p < 0.001-0.038$) on most of the variables studied as soon as the infant and cadet categories were compared with any of the other categories, with the exception of the BMI variable which showed no differences in any comparison ($p = 0.236-1.000$).

Table 4 shows the distribution (%) of the degree of adherence to MD, calculated by the score obtained in the KIDMED questionnaire, according to category. No significant difference was found between categories ($p = 0.204$). However, the infant and senior categories had the fewest players with poor adherence (2.9% and 7.7%, respectively), while

Romero-García et al. (2022), *PeerJ*, DOI 10.7717/peerj.14329

**Table 1 Descriptive data (mean ± standard deviation) and differences according to category, including main effects of the covariate age and its intercept.**

| Variable | Category (Mean ± SD) | | | | Category | | | Age | | | Category*age | | |
|---|---|---|---|---|---|---|---|---|---|---|---|---|---|
| | Infant (*n* = 35) | Cadet (*n* = 46) | Junior (*n* = 26) | Senior (*n* = 26) | F | *p* | $\eta^2_p$ | F | *p* | $\eta^2_p$ | F | *p* | $\eta^2_p$ |
| Body Mass (kg) | 52.80 ± 12.15 | 61.94 ± 12.79 | 72.69 ± 11.08 | 77.11 ± 15.78 | 21.72 | **0.000** | 0.336 | 43.721 | **0.000** | 0.250 | 15.375 | **0.000** | 0.325 |
| Height (m) | 1.59 ± 0.10 | 1.72 ± 0.06 | 1.77 ± 0.06 | 1.75 ± 0.05 | 36.81 | **0.000** | 0.462 | 37.631 | **0.000** | 0.223 | 27.125 | **0.000** | 0.459 |
| BMI (kg/m2) | 20.53 ± 3.68 | 20.90 ± 3.54 | 23.05 ± 3.22 | 24.94 ± 4.24 | 9.424 | **0.000** | 0.180 | 21.958 | **0.000** | 0.144 | 6.530 | **0.000** | 0.169 |
| FM (%) | 21.44 ± 9.67 | 14.52 ± 7.04 | 15.07 ± 7.01 | 17.56 ± 9.34 | 5.250 | **0.002** | 0.109 | 1.287 | 0.259 | 0.100 | 4.013 | **0.004** | 0.111 |
| $\sum$3-skinfolds (mm) | 46.79 ± 19.60 | 32.38 ± 17.88 | 33.06 ± 16.74 | 41.18 ± 26.15 | 4.259 | **0.007** | 0.090 | 0.266 | 0.607 | 0.002 | 3.196 | **0.015** | 0.091 |
| MM (kg) | 22.31 ± 4.86 | 26.74 ± 5.87 | 32.64 ± 4.79 | 34.68 ± 6.40 | 31.706 | **0.000** | 0.424 | 68.840 | **0.000** | 0.344 | 23.249 | **0.000** | 0.421 |
| $\sum$3-girths sum corrected (mm) | 98.19 ± 10.82 | 105.21 ± 12.17 | 115.76 ± 8.97 | 118.89 ± 13.49 | 22.990 | **0.000** | 0.348 | 49.539 | **0.000** | 0.274 | 16.631 | **0.000** | 0.342 |
| KM | 7.40 ± 1.98 | 6.37 ± 2.21 | 6.92 ± 2.46 | 6.38 ± 2.15 | 1.773 | 0.156 | 0.040 | 0.328 | 0.568 | 0.003 | 1.446 | 0.223 | 0.043 |
| MB Throw (m) | 4.39 ± 0.99 | 6.26 ± 1.13 | 7.66 ± 1.24 | 7.60 ± 1.14 | 58.408 | **0.000** | 0.576 | 78.554 | **0.000** | 0.375 | 43.773 | **0.000** | 0.578 |
| T-Half Test (s) | 7.05 ± 0.71 | 6.55 ± 0.64 | 6.01 ± 0.53 | 6.45 ± 0.47 | 14.872 | **0.000** | 0.257 | 8.480 | **0.004** | 0.061 | 11.428 | **0.000** | 0.263 |
| SJ (cm) | 18.94 ± 6.16 | 25.45 ± 4.72 | 30.28 ± 4.97 | 24.82 ± 6.61 | 21.423 | **0.000** | 0.333 | 10.423 | **0.002** | 0.074 | 16.254 | **0.000** | 0.337 |
| CMJ (cm) | 20.85 ± 6.26 | 29.87 ± 5.20 | 34.77 ± 5.90 | 30.79 ± 6.43 | 31.790 | **0.000** | 0.425 | 22.293 | **0.000** | 0.145 | 23.978 | **0.000** | 0.428 |
| Sprint (s) | 4.76 ± 0.34 | 4.29 ± 0.32 | 4.09 ± 0.22 | 4.31 ± 0.36 | 25.748 | **0.000** | 0.375 | 9.563 | **0.002** | 0.068 | 20.999 | **0.000** | 0.396 |
| YoYo distance (m) | 757 ± 453 | 1142 ± 420 | 1083 ± 415 | 1091 ± 452 | 5.911 | **0.001** | 0.121 | 3.603 | 0.060 | 0.027 | 4.517 | **0.002** | 0.124 |
| YoYo VO$_2$ max (mL/min/kg) | 42.75 ± 3.80 | 45.99 ± 3.53 | 45.50 ± 3.49 | 45.56 ± 3.79 | 5.911 | **0.001** | 0.121 | 3.603 | 0.060 | 0.027 | 4.517 | **0.002** | 0.124 |

**Notes.**

SD, standard deviation; BMI, Body Mass Index; FM, Fat mass; $\sum$3-skinfolds, Sum of 3-skinfolds; MM, Muscle mass; $\sum$3-girths sum corrected, Sum of corrected girths; KM, Kidmed; MB, Medicine ball; SJ, Squad Jump; CMJ, Counter Movement Jump.

Values in bold are statistically significant ($p < 0.05$).

**Table 2  *Post hoc* comparison between categories with significant differences in ANCOVA analysis for kinanthropometric characteristics.**

| Variable | Group comparison | | Model | | | | | |
|---|---|---|---|---|---|---|---|---|
| | | | Category | | | Category*age | | |
| | | | Mean difference ± SD | p | 95% CI | Mean difference ± SD | p | 95% CI |
| Body mass (kg) | I | C | −9.14 ± 2.91 | **0.012** | −16.93 to −1.34 | −8.80 ± 3.17 | **0.038** | −17.29 to −0.31 |
| | I | J | −19.89 ± 3.36 | **0.000** | −28.88 to −10.89 | −18.99 ± 4.71 | **0.001** | −31.64 to −6.35 |
| | I | S | −24.31 ± 3.36 | **0.000** | −33.30 to −15.31 | −22.73 ± 6.72 | **0.006** | −40.75 to −4.72 |
| | C | J | −10.75 ± 3.18 | **0.006** | −19.27 to −2.22 | −10.19 ± 3.80 | **0.050** | −20.38 to −0.00 |
| | C | S | −15.17 ± 3.18 | **0.000** | −23.69 to −6.65 | −13.93 ± 5.58 | 0.083 | −28.89 to 1.029 |
| | J | S | −4.42 ± 3.59 | 1.000 | −14.05 to 5.21 | −3.740 ± 4.39 | 1.000 | −15.53 to 8.05 |
| Height (m) | I | C | −0.12 ± 0.02 | **0.000** | −0.16 to −0.07 | −0.12 ± 0.02 | **0.000** | −0.16 to −0.07 |
| | I | J | −0.17 ± 0.02 | **0.000** | −0.23 to −0.12 | −0.17 ± 0.03 | **0.000** | −0.24 to −0.09 |
| | I | S | −0.15 ± 0.02 | **0.000** | −0.21 to −0.10 | −0.14 ± 0.04 | **0.001** | −0.24 to −0.04 |
| | C | J | −0.06 ± 0.018 | **0.009** | −0.11 to −0.10 | −0.05 ± 0.02 | 0.079 | −0.11 to 0.00 |
| | C | S | −0.04 ± 0.08 | 0.223 | −0.08 to −0.01 | −0.03 ± 0.03 | 1.000 | −0.11 to 0.06 |
| | J | S | 0.02 ± 0.02 | 1.000 | −0.03 to 0.07 | 0.03 ± 0.02 | 1.000 | −0.04 to 0.09 |
| BMI (kg/m2) | I | C | −0.37 ± 0.83 | 1.000 | −2.59 to 1.84 | −0.29 ± 0.90 | 1.000 | −2.69 to 2.14 |
| | I | J | −2.52 ± 0.95 | 0.056 | −5.08 to 0.04 | −2.276 ± 1.34 | 0.554 | −5.873 to 1.32 |
| | I | S | −4.41 ± 0.95 | **0.000** | −6.97 to −1.85 | −2.276 ± 1.34 | 0.236 | −9.11 to 1.14 |
| | C | J | −2.15 ± 0.90 | 0.115 | −4.57 to 0.28 | −1.99 ± 1.08 | 0.404 | −4.89 to 0.90 |
| | C | S | −4.04 ± 0.90 | **0.000** | −6.47 to −1.61 | −3.70 ± 1.59 | 0.128 | −7.96 to 0.55 |
| | J | S | −1.89 ± 1.02 | 0.400 | −4.63 to 0.85 | −1.71 ± 1.25 | 1.000 | −5.01 to 1.65 |
| FM (%) | I | C | 6.92 ± 1.85 | **0.002** | 1.95 to 11.89 | 6.67 ± 2.02 | **0.008** | 1.25 to 12.08 |
| | I | J | 6.36 ± 2.14 | **0.021** | 0.63 to 12.10 | 5.69 ± 3.01 | 0.363 | −2.37 to 13.76 |
| | I | S | 3.88 ± 2.14 | 0.433 | −1.85 to 9.62 | 2.70 ± 4.29 | 1.000 | −8.79 to 14.19 |
| | C | J | −0.55 ± 2.03 | 1.000 | −5.99 to 4.88 | −0.93 ± 2.42 | 1.000 | −7.47 to 5.53 |
| | C | S | −3.04 ± 2.03 | 0.821 | −8.47 to 2.40 | −3.96 ± 3.56 | 1.000 | −13.51 to 5.58 |
| | J | S | −2.48 ± 2.29 | 1.000 | −8.63 to 3.66 | −2.99 ± 2.80 | 1.000 | −10.51 to 4.52 |
| ∑3-skinfolds (mm) | I | C | 14.51 ± 4.48 | **0.009** | 2.49 to 26.52 | 14.31 ± 4.89 | **0.024** | 1.21 to 27.41 |
| | I | J | 13.73 ± 5.17 | 0.054 | −0.14 to 27.59 | 13.19 ± 7.27 | 0.433 | −6.30 to 32.69 |
| | I | S | 5.61 ± 5.17 | 1.000 | −8.25 to 19.48 | 4.67 ± 10.37 | 1.000 | −23.11 to 32.45 |
| | C | J | −0.78 ± 4.90 | 1.000 | −13.92 to 12.36 | −1.12 ± 5.86 | 1.000 | −16.83 to 14.59 |
| | C | S | −8.89 ± 4.90 | 0.432 | −22.03 to 4.25 | −9.64 ± 8.61 | 1.000 | −32.71 to 13.43 |
| | J | S | −8.11 ± 5.54 | 0.875 | −22.97 to 6.74 | −8.52 ± 6.78 | 1.000 | −26.70 to 9.66 |
| MM (kg) | I | C | −4.43 ± 1.24 | **0.003** | −7.76 to −1.01 | −3.85 ± 1.35 | **0.030** | −7.46 to −0.23 |
| | I | J | −10.33 ± 1.43 | **0.000** | −14.17 to −6.49 | −8.76 ± 2.01 | **0.000** | −14.14 to −3.39 |
| | I | S | −12.37 ± 1.43 | **0.000** | −16.21 to −8.53 | −9.61 ± 2.86 | **0.006** | −17.27 to −1.95 |
| | C | J | −5.90 ± 1.36 | **0.000** | −9.54 to −2.26 | −4.92 ± 1.62 | **0.017** | −9.25 to −0.59 |
| | C | S | −7.93 ± 1.36 | **0.000** | −11.58 to −4.30 | −5.76 ± 2.37 | 0.099 | −12.12 to 0.59 |
| | J | S | −2.04 ± 1.53 | 1.000 | −6.15 to 2.08 | −0.84 ± 1.87 | 1.000 | −5.86 to 4.17 |

**Table 2** (*continued*)

| Variable | Group comparison | | Model | | | | | |
|---|---|---|---|---|---|---|---|---|
| | | | Category | | | Category*age | | |
| | | | Mean difference ± SD | p | 95% CI | Mean difference ± SD | p | 95% CI |
| | I | C | −7.01 ± 2.47 | **0.032** | −13.63 to −0.39 | −6.26 ± 2.69 | 0.129 | −13.47 to 0.95 |
| $\sum$3-girths corrected sum (mm) | I | J | −17.56 ± 2.85 | **0.000** | −25.20 to −9.92 | −15.55 ± 4.00 | **0.001** | −26.27 to −4.83 |
| | I | S | −20.70 ± 2.85 | **0.000** | −28.34 to −13.05 | −17.15 ± 5.70 | **0.019** | −32.44 to −1.87 |
| | C | J | −10.55 ± 2.70 | **0.001** | −17.79 to −3.30 | −9.29 ± 3.22 | **0.028** | −17.93 to −0.65 |
| | C | S | −13.68 ± 2.70 | **0.000** | −20.92 to −6.44 | −10.89 ± 4.74 | 0.138 | −23.59 to 1.79 |
| | J | S | −3.15 ± 3-05 | 1.000 | −11.32 to 5.05 | −1.60 ± 3.73 | 1.000 | −11.60 to 8.39 |

**Notes.**

SD, standard deviation; BMI, Body Mass Index; FM, Fat mass; $\sum$3-skinfolds, Sum of 3-skinfolds; MM, Muscle mass; $\sum$3 corrected PR, Sum of corrected girths.
Values in bold are statistically significant ($p < 0.05$).

the infant and junior categories had the most players with good adherence (54.3% and 50.0%, respectively). With respect to the total, only 10.5% had a poor level of adherence, while the predominant level was medium with 47.4%.

All figures show the bivariate correlations between anthropometric characteristics and physical test results, and between KIDMED score and both. The KIDMED questionnaire showed low negative correlations with body mass, height, BMI, MM and $\sum$3-girths sum corrected ($r = -0.190$ to $-0.236$; $p < 0.006 - 0.019$), whereas it did not correlate with any variable from the the physical tests ($r = -0.148$ to $0.061$; $p = 0.088 - 0.956$) (Fig. 1). Upper limb power, assessed with the medicine ball throw, showed high positive correlations with age, body mass, height, BMI, MM and $\sum$3-girths sum corrected ($r = 0.535$ to $0.784$; $p < 0.001$) (Fig. 2). In the agility test (T-Half Test), low to moderate negative correlations were found between the time taken to complete the run, with age, height, MM and $\sum$3-girths sum corrected ($r = -0.397$ to $-0.179$; $p < 0.001 - 0.039$), and moderate positive correlations with both FM and $\sum$3-skinfolds ($r = 0.387$ to $0.389$; $p < 0.001 - 0.03$) (Fig. 3). The SJ (Fig. 4) and CMJ (Fig. 5) showed low to moderate positive correlations with age, height, and MM ($r = 0.180$ to $0.489$; $p < 0.001 - 0.038$), and low to moderate negative correlations with FM and the $\sum$3-skinfolds ($r = -0.427$ to $-0.537$; $p < 0.001$). In addition, the CMJ was also positively correlated with a low association with the $\sum$3-girths sum corrected ($r = 0.197$; $p < 0.023$). In speed, with a longer time in sprint indicating a worse performance, low to moderate negative correlations were observed with both age and height ($r = -0.261$ to $-0.445$; $p < 0.001 - 0.002$), and high positive correlations with both FM and $\sum$3-skinfolds ($r = 0.636$ to $0.645$; $p < 0.001$) (Fig. 6). Regarding endurance, both distance (Fig. 7) and VO$_2$ max (Fig. 8), low to high negative correlations were observed with BMI, FM and $\sum$3-skinfolds ($r = -0.540$ to $-0.294$; $p < 0.001$), and low positive correlation with height ($r = 0.219$; $p < 0.011$).

Table 5 shows the multiple linear regression analysis, as well as the resulting predictive equations for physical capacity and KIDMED score. With the exception of the latter, where the independent variable included only explained 7% of its variability ($p = 0.006$), the anthropometric characteristics included in each model explained 32% to 72% of the

**Table 3** *Post hoc* comparison between categories with significant differences in the ANCOVA analysis for physical capacity results.

| Variable | Group comparison | | Model | | | | | |
|---|---|---|---|---|---|---|---|---|
| | | | Category | | | Category*age | | |
| | | | Mean difference ± SD | *p* | 95% CI | Mean difference ± SD | *p* | 95% CI |
| MB Throw (m) | I | C | −1.87 ± 0.25 | **0.000** | −2.54 to −1.19 | −1.74 ± 0.27 | **0.000** | −2.47 to −1.01 |
| | I | J | −3.26 ± 0.29 | **0.000** | −4.04 to −2.49 | −2.91 ± 0.40 | **0.000** | −3.99 to −1.83 |
| | I | S | −3.21 ± 0.29 | **0.000** | −3.99 to −2.43 | −2.59 ± 0.58 | **0.000** | −4.14 to −1.05 |
| | C | J | −1.39 ± 0.27 | **0.000** | −2.13 to −0.66 | −1.17 ± 0.33 | **0.003** | −2.05 to −0.30 |
| | C | S | −1.34 ± 0.27 | **0.000** | −2.08 to −0.60 | −0.85 ± 0.48 | 0.462 | −2.14 to 0.43 |
| | J | S | 0.05 ± 0.31 | 1.000 | −0.78 to 0.88 | 0.32 ± 0.38 | 1.000 | −0.69 to 1.33 |
| T-Half Test (s) | I | C | 0.50 ± 0.14 | **0.002** | 0.13 to 0.87 | 0.57 ± 0.15 | **0.001** | 0.17 to 0.97 |
| | I | J | 1.04 ± 0.16 | **0.000** | 0.62 to 1.47 | 1.22 ± 0.22 | **0.000** | 0.63 to 1.81 |
| | I | S | 0.60 ± 0.16 | **0.001** | 0.18 to 1.02 | 0.91 ± 0.32 | **0.026** | 0.069 to 1.76 |
| | C | J | 0.54 ± 0.15 | **0.003** | 0.14 to 0.94 | 0.65 ± 0.18 | **0.002** | 0.17 to 1.13 |
| | C | S | 0.10 ± 0.15 | 1.000 | −0.30 to 0.50 | 0.35 ± 0.26 | 1.000 | −0.36 to 1.05 |
| | J | S | −0.44 ± 0.17 | 0.063 | −0.89 to 0.01 | −0.30 ± 0.21 | 0.849 | −0.86 to 0.25 |
| SJ (cm) | I | C | −6.51 ± 1.25 | **0.000** | −9.86 to −3.16 | −6.54 ± 1.36 | **0.000** | −10.19 to −2.89 |
| | I | J | −11.34 ± 1.44 | **0.000** | −15.20 to −7.47 | −11.42 ± 2.03 | **0.000** | −16.85 to −5.99 |
| | I | S | −5.87 ± 1.44 | **0.000** | −9.74 to −2.01 | −6.01 ± 2.89 | 0.236 | −13.76 to 1.73 |
| | C | J | −4.83 ± 1.37 | **0.003** | −8.49 to −1.18 | −4.88 ± 1.63 | **0.020** | −9.26 to −0.502 |
| | C | S | 0.64 ± 1.37 | 1.000 | −3.02 to 4.29 | 0.52 ± 2.39 | 1.000 | −5.90 to 6.95 |
| | J | S | 5.47 ± 1.54 | **0.003** | 1.33 to 9.60 | 5.40 ± 1.89 | **0.030** | 0.34 to 10.47 |
| CMJ (cm) | I | C | −9.02 ± 1.32 | **0.000** | −12.55 to −5.49 | −9.04 ± 1.44 | **0.000** | −12.89 to −5.19 |
| | I | J | −13.92 ± 1.52 | **0.000** | −17.99 to −9.85 | −13.98 ± 2.14 | **0.000** | −19.71 to −8.25 |
| | I | S | −9.94 ± 1.52 | **0.000** | −14.02 to −5.87 | −10.04 ± 3.05 | **0.008** | −18.21 to −1.87 |
| | C | J | −4.90 ± 1.44 | **0.005** | −8.76 to −1.04 | −4.94 ± 1.72 | **0.029** | −9.56 to −0.32 |
| | C | S | −0.92 ± 1.44 | 1.000 | −4.79 to 2.94 | −1.00 ± 2.53 | 1.000 | −7.78 to 5.78 |
| | J | S | 3.98 ± 1.63 | 0.096 | −0.39 to 8.34 | 3.94 ± 1.99 | 0.303 | −1.41 to 9.28 |
| Sprint (s) | I | C | 0.48 ± 0.07 | **0.000** | 0.28 to 0.66 | 0.53 ± 0.08 | **0.000** | 0.33 to 0.73 |
| | I | J | 0.67 ± 0.08 | **0.000** | 0.45 to 0.89 | 0.84 ± 0.11 | **0.000** | 0.53 to 1.14 |
| | I | S | 0.46 ± 0.08 | **0.000** | 0.24 to 0.68 | 0.75 ± 0.16 | **0.000** | 0.32 to 1.18 |
| | C | J | 0.20 ± 0.08 | 0.062 | −0.01 to 0.41 | 0.31 ± 0.09 | **0.006** | 0.06 to 0.55 |
| | C | S | −0.01 ± 0.08 | 1.000 | −0.22 to 0.19 | 0.22 ± 0.13 | 0.134 | −0.14 to 0.58 |
| | J | S | −0.21 ± 0.09 | 0.108 | −0.44 to 0.02 | −0.08 ± 0.11 | 1.000 | −0.37 to 0.20 |
| YoYo (m) | I | C | −385 ± 97 | **0.001** | −646 to −124 | −457 ± 128 | **0.003** | −802 to −113 |
| | I | J | −326 ± 112 | **0.026** | −627 to −25 | −394 ± 188 | 0.228 | −898 to 110 |
| | I | S | −334 ± 112 | **0.021** | −635 to −33 | −404 ± 254 | 0.680 | −1084 to 275 |
| | C | J | 59 ± 106 | 1.000 | −227 to 344 | 63 ± 133 | 1.000 | −294 to 421 |
| | C | S | 51 ± 106 | 1.000 | −234 to 336 | 53 ± 190 | 1.000 | −457 to 563 |
| | J | S | 8 ± 120 | 1.000 | −330 to 315 | −10 ± 132 | 1.000 | −364 to 344 |

**Table 3** (*continued*)

| Variable | Group comparison | | Model | | | | | |
|---|---|---|---|---|---|---|---|---|
| | | | **Category** | | | **Category*age** | | |
| | | | Mean difference ± SD | *p* | 95% CI | Mean difference ± SD | *p* | 95% CI |
| YoYo VO₂ max (mL/min/kg) | I | C | −3.23 ± 0.82 | **0.001** | −5.43 to −1.04 | −3.84 ± 1.08 | **0.003** | −6.74 to −0.95 |
| | I | J | −2.74 ± 0.94 | **0.026** | −5.27 to −0.21 | −3.31 ± 1.58 | 0.228 | −7.54 to 0.92 |
| | I | S | −2.81 ± 0.94 | **0.021** | −5.34 to −0.28 | −3.40 ± 2.13 | 0.680 | −9.10 to 2.31 |
| | C | J | 0.49 ± 0.89 | 1.000 | −1.90 to 2.89 | 0.53 ± 1.12 | 1.000 | −2.47 to 3.53 |
| | C | S | 0.43 ± 0.89 | 1.000 | −1.97 to 2.83 | 0.45 ± 1.60 | 1.000 | −3.84 to 4.73 |
| | J | S | −0.06 ± 1.01 | 1.000 | −2.77 to 2.65 | −0.08 ± 1.11 | 1.000 | −3.06 to 2.89 |

**Notes.**

SD, standard deviation; MB, Medicine ball; SJ, Squad Jump; CMJ, Counter Movement Jump.
Values in bold are statistically significant ($p < 0.05$).

**Table 4** Distribution (%) of the level of adherence to the Mediterranean diet according to category.

| Adhesion | Infant (n = 35) | Cadet (n = 46) | Junior (n = 26) | Senior (n = 26) | Total (n = 133) | X²/p |
|---|---|---|---|---|---|---|
| Good (%) | 54.3 | 32.6 | 50.0 | 34.6 | 42.1 | X² = 8.501; p = 0.204 |
| Average (%) | 42.9 | 52.2 | 34.6 | 57.7 | 47.4 | |
| Poor (%) | 2.9 | 15.2 | 15.4 | 7.7 | 10.5 | |

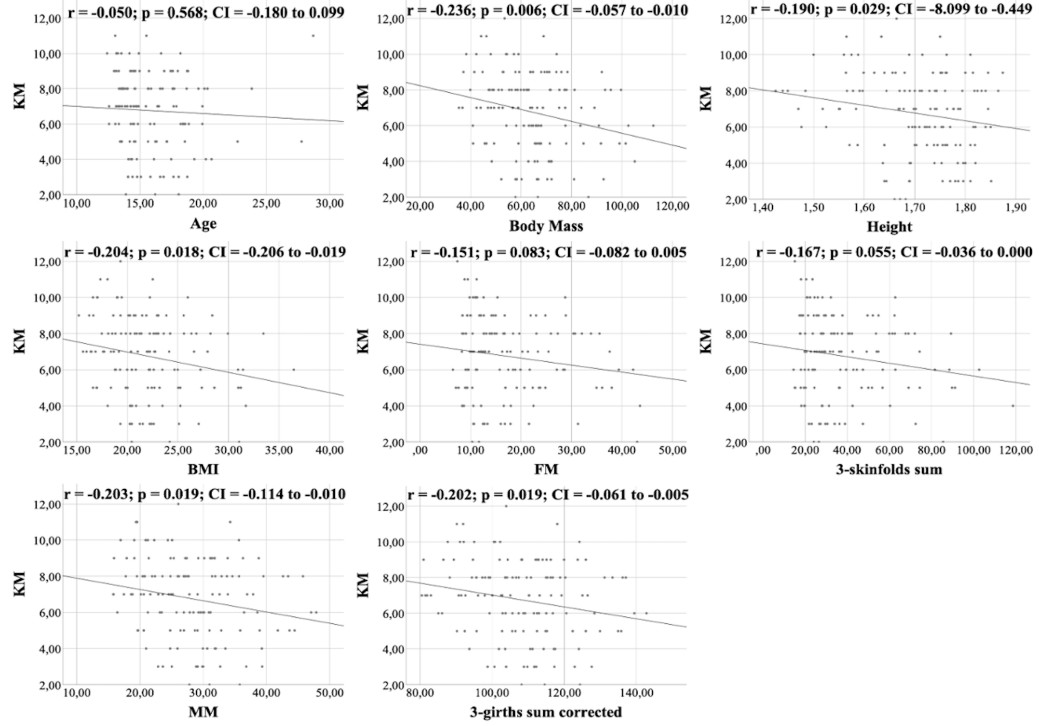

**Figure 1** Correlations between KIDMED score and kinanthropometric characteristics.

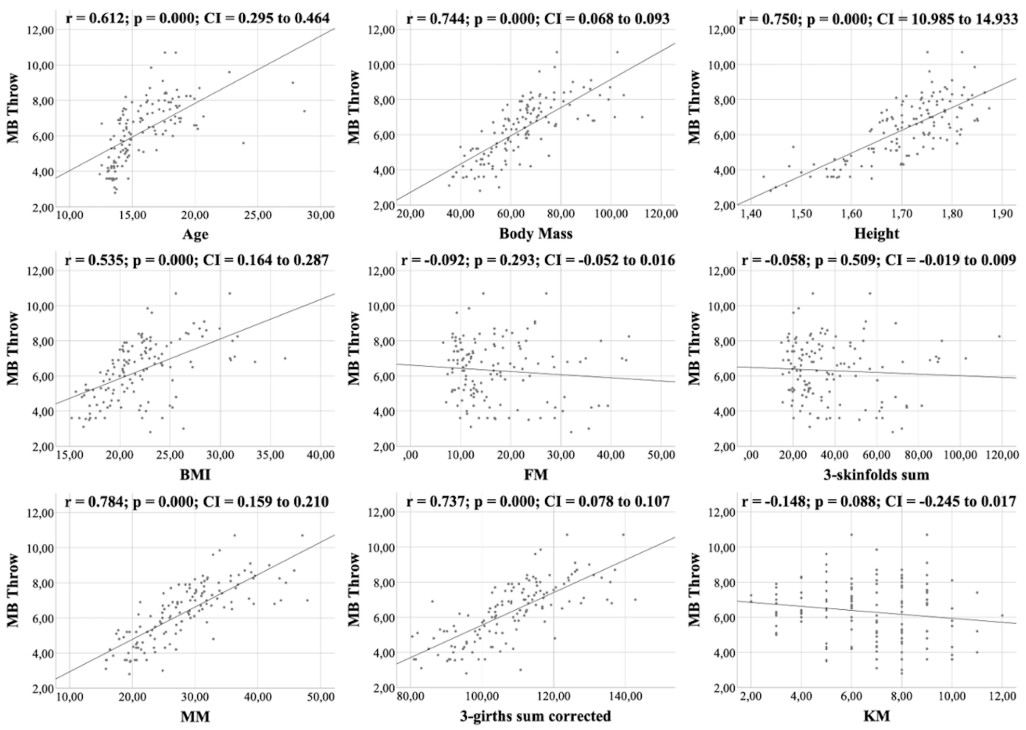

**Figure 2  Correlations between medicine ball throwing, kinanthropometric characteristics and KIDMED score.**

variability of the physical test results ($p < 0.001$). The most determinant anthropometric characteristics were MM and FM.

## DISCUSSION

The main objective was to observe if there were any differences in the anthropometric characteristics, adherence to MD and physical condition as a function of the sports category in handball. It was found that there were significant differences between the categories compared, both in most of the anthropometric characteristics, and in the results of the physical tests; while no significant differences were found between the groups studied in the distribution of the degree of adherence to the MD. In this study, it was found that when the players were part of a category whose age range was older, most of the anthropometric characteristics also increased, with the exception of FM and $\sum$3-skinfolds, and that the results in the physical tests were better. *Hermassi et al. (2020)* also found differences in all kinanthropometric parameters between age groups in handball players, with the exception of body fat percentage. However, for the physical tests, they found no significant differences, with the exception of upper limb power in the medicine ball throw test. This could possibly be due to the fact that their study population were young players aged 10 to 12 years old, so the maturation process, with all the hormonal changes that it entails, had not yet begun. *Molina-López et al. (2020)* found the same results as the present work for both anthropometric characteristics and physical test results in handball

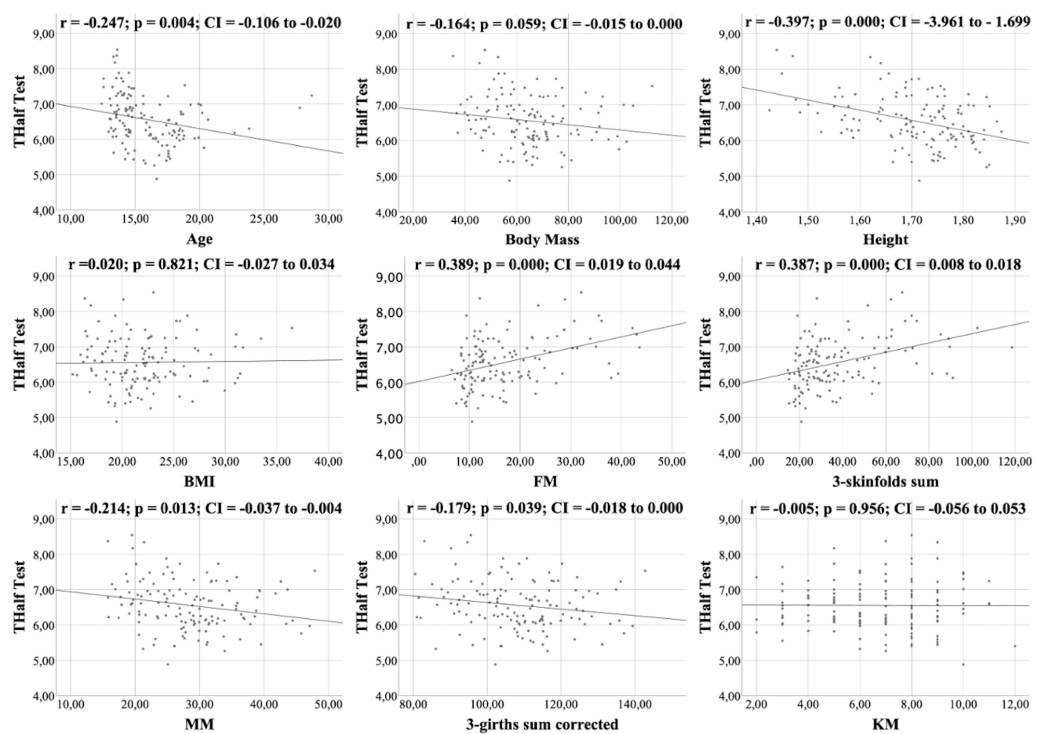

**Figure 3  Correlations between T-Half test, kinanthropometric characteristics and KIDMED score.**

players. Moreover, as in this study, the differences between categories were reduced or even not observed when comparing groups further away from puberty, which could be due to the stabilization of the maximum growth peak, and the end of the maturation process (*Albaladejo-Saura et al., 2021*). Therefore, as players grow older, their body composition changes (*Albaladejo-Saura et al., 2022*), as can be hypothesized by analyzing the results of the present investigation, allowing in most cases, to have a better performance in physical fitness tests (*Albaladejo-Saura et al., 2021*). However, this improvement in body composition and performance is also influenced by other reasons, such as the level of play or hours of training (*Hammami et al., 2019*), so more research is still lacking for a better understanding of the factors that affect body composition and performance of players.

Another objective of this research was to determine the degree of adherence to the MD of handball players. In this line, the findings that were found indicated that the athletes have a medium and good level of adherence to the MD (47.4% and 42.1%, respectively). These findings are similar to those found in several studies, where it was also described that the predominant degrees of adherence were medium or good (*Vaquero-Cristóbal et al., 2018*; *Manzano-Carrasco et al., 2020a*; *Martínez-Rodríguez et al., 2021*). Along these lines, (*Manzano-Carrasco et al., 2020a*) evaluated athletes of both sexes, and found that 57.6% of men and 59.8% of women had a medium level of adherence. (*Martínez-Rodríguez et al., 2021*), who evaluated both male and female beach handball players, found that 66.0% of men and 76.0% of women also had a medium level of adherence. *Vaquero-Cristóbal et al.*

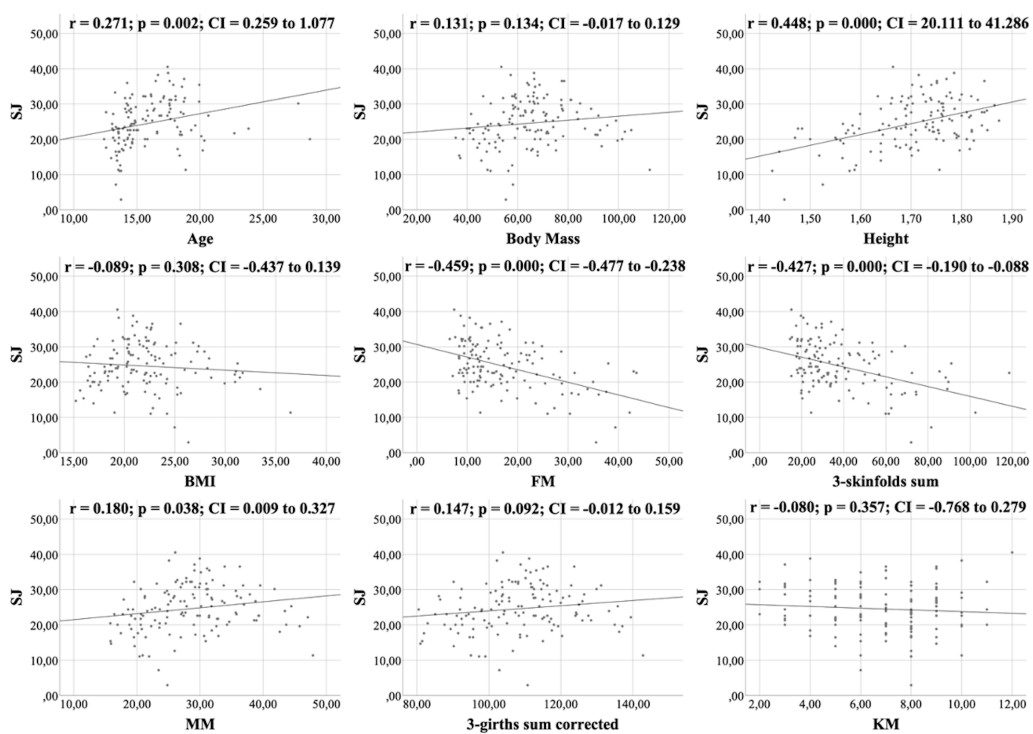

**Figure 4** Correlations between the SJ test, kinanthropometric characteristics and the KIDMED score.

*(2018)*, evaluated elite cadet male canoeists, and found that more than half of the sample
had a good degree of adherence to the MD. All of these results are similar to those found
in the AdolesHealth study (*Galan-Lopez et al., 2019*), which evaluated 1,717 adolescents of
both sexes, from different European cities and schools, and which found that 54.92% had
an average adherence to the Mediterranean diet. However, that the dietary pattern of young
athletes is the same as the general adolescent population does not mean that it is optimal,
as athletes have different requirements to maintain performance during the competitive
season (*Thomas, Erdman & Burke, 2016*). This is an important point for future research.

On the other hand, and despite the fact that in our sample the predominant levels of
adherence were good and medium, no significant differences were found in the distribution
of the level of MD adherence. *Manzano-Carrasco et al. (2020b)*, after evaluating young
soccer players, also found no significant differences in the KIDMED questionnaire score.
This could be due to the fact that the MD is being substituted by foods characteristic of
other cultures, mainly affecting adolescents (*Rubio-Árias et al., 2015*). However, there is
a lack of more research that delves deeper into this. Another aspect to highlight is that
the degree of adherence to the MD has not been correlated with the results of physical
tests, but with some anthropometric characteristics. *Vaquero-Cristóbal et al. (2018)* found
no relationship between adherence to the MD and anthropometric parameters of male
canoeists. *Martínez-Rodríguez et al. (2021)* concluded that having a good MD adherence is
not enough for obtaining better results in the performance tests in male and female beach

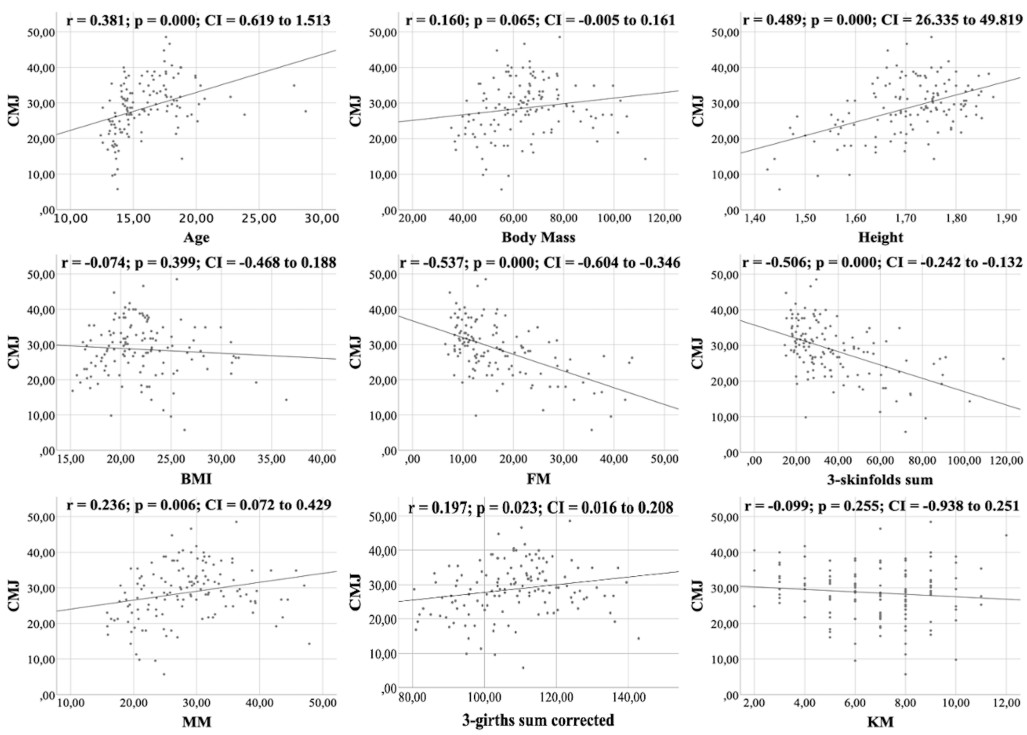

**Figure 5** Correlations between the CMJ test, kinanthropometric characteristics and the KIDMED score.

handball players. It has been suggested that this could be due to the homogeneity in the adherence to the MD shown by athletes (*Vaquero-Cristóbal et al., 2018*). Even then, more research is still lacking, both in handball and in other sports, for an in-depth analysis of a sufficient sample size, of whether having a good adherence to the MD or specific eating habits influences performance and body composition or anthropometric characteristics.

The last objective of this research was to know if there was a relationship between anthropometric characteristics and sports performance. Along this line, the findings indicate that both FM percentage and MM are important variables that describe the variability of the results of the physical tests. More specifically, a low percentage of FM, a good development of MM, or both, improve the results of the physical tests. The findings found regarding FM are in agreement with those found by *Hermassi et al. (2021)*, who evaluated obese and non-obese young male handball players, and concluded that the percentage of FM negatively affected speed, vertical jump and aerobic capacity, three basic and necessary skills for a handball player. In another study by *Hermassi, Laudner & Schwesig (2019)* conducted with young male handball players, the authors found that, with the exception of speed, there was a negative association between FM and the vertical jump, SJ and CMJ performance tests. However, in another study by *Hammami et al. (2019)* they found that FM was not related to performance measurements, with the exception of vertical and horizontal jumps. *Moncef et al. (2012)* also found no correlations between body fat and the vertical jump, SJ and CMJ tests, nor with the endurance capacity of the

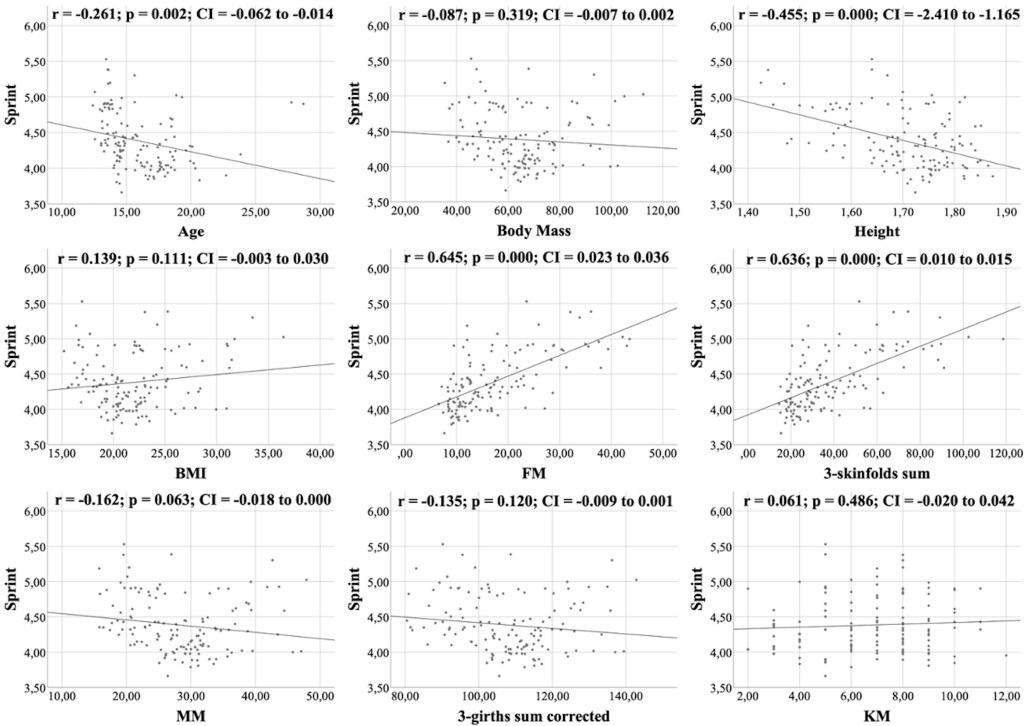

**Figure 6** Correlations between sprint, kinanthropometric characteristics and KIDMED score.

players. *Molina-López et al. (2020)*, after evaluating young handball players, concluded that MM analysis proved to be more useful than FM assessment in determining performance. These differences could be due to the heterogeneity in the competitive level of the handball players included in the studies, so future lines of research should analyze the specific weight of FM and MM according to the level of the players.

Regarding the limitations of this study, the main one was the sample size. It should also be taken into account that the KIDMED questionnaire does not report the amount of food ingested. Furthermore, as the study was cross-sectional, no MD-based nutritional interventions, which could have improved body composition and performance, could have been performed. As for future research, this study should be replicated with women. Another interesting aspect would be to be able to perform a nutritional intervention based on the MD and to evaluate whether players improve their body composition and performance. In this way, it would be possible to analyze in depth, whether having an excellent association with the MD is related to better performance.

As possible practical applications both for handball professionals (sports federations, clubs, coaches, physical trainers, *etc.*) and for the players themselves, it should be noted that this study shows that the plastic components of body composition (adiposity and muscle development) influence the physical condition tests related to handball performance, so that performance in this sport could be improved by modifying these components. In this way, the anthropometric characteristics of the players should be monitored throughout the

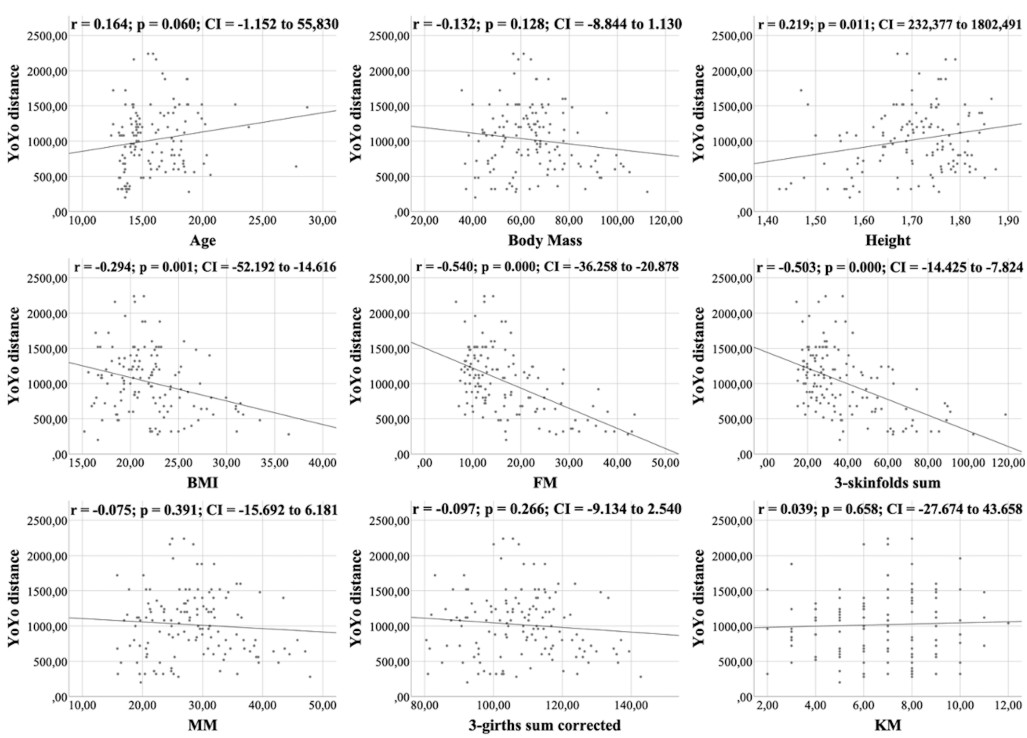

**Figure 7 Correlations between YOYO test distance, kinanthropometric characteristics and KIDMED score.**

season in order to be able to individualize the nutritional and training plans to the needs of each player in the search for the optimization of sports performance (*Martínez-Rodríguez et al., 2020*; *Ghobadi et al., 2013*). Another practical application of the present study is that taller players have a competitive advantage in physical fitness tests related to handball performance. Because of the influence of maturation on this parameter (*Albaladejo-Saura et al., 2021*), the selection of players with early maturation during these ages could bring better short-term performance (*Albaladejo-Saura et al., 2022*), although it is not clear that this is an advantage when the maturation process is completed and the characteristics are equalized (*Albaladejo-Saura et al., 2021*). On the other hand, there is no great awareness on the part of the players of the importance of having a healthy diet in the search for the optimization of their sports performance, especially in the pubertal stage, so it would be necessary to carry out informative programs from an early age to increase awareness among athletes on this issue.

## CONCLUSIONS

In conclusion, there are differences according to category in most of the kinanthropometric and physical condition variables, but not in the level of adherence to the MD, highlighting that as the players were from a category with an older age range, the anthropometric characteristics and the results of the physical tests were better. In addition, most of the subjects had a medium and good level of adherence to the MD, and very few players showed

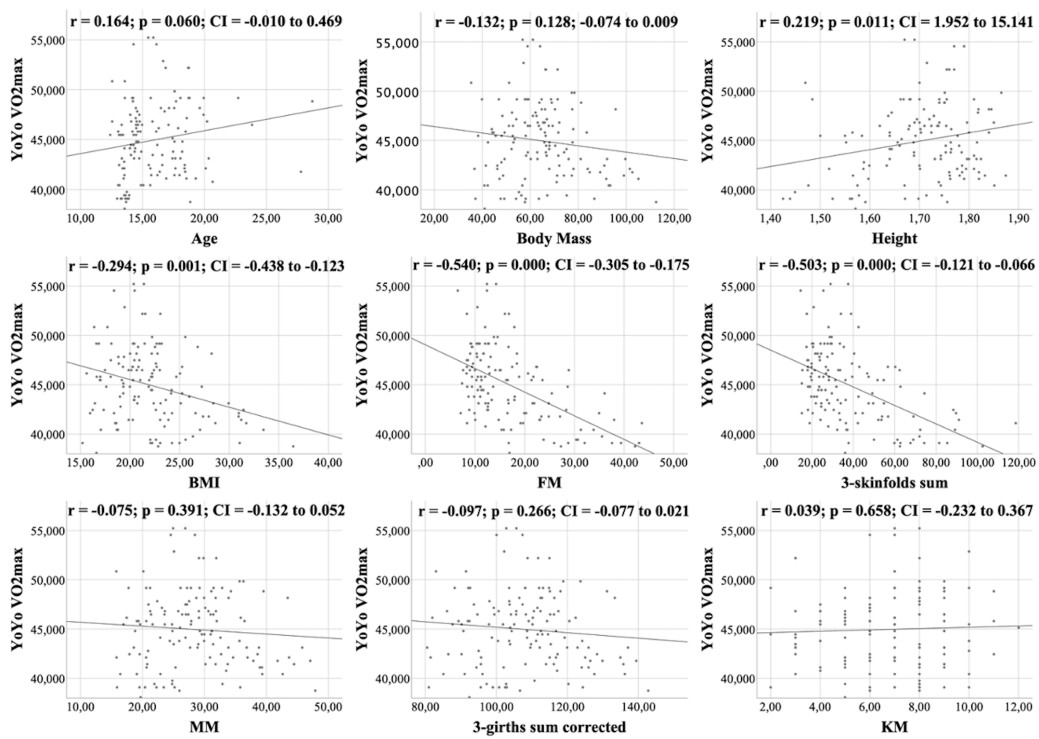

**Figure 8** Correlations between YOYO test VO$_2$ max, kinanthropometric characteristics and KIDMED score.

a poor adherence. However, this level of adherence was not associated to the results of the physical tests. Nevertheless, a relationship between anthropometric characteristics and sports performance was observed; having a low percentage of FM, having a good development of MM, or both, improves the results of the physical tests.

# ACKNOWLEDGEMENTS

The authors would like to thank the clubs and coaches who made this study possible, as well as the volunteer players who participated, for their collaboration. We would also like to thank the researchers who helped in the measurement sessions of the Prevention of Sports Injuries group of the Catholic University San Antonio of Murcia and the Nursing department, Food and Nutrition Research Group (ALINUT) of the University of Alicante, without whom this research would not have been possible. This study is part of the doctoral thesis of David Romero-García.

## Funding

The authors received no funding for this work.

**Table 5  Analysis and regression models of performance in the different physical capacities.**

| Variable | Analysis | $R^2$ | $p$ | Independent variables included | Standard coefficient (ß) | Equation |
|---|---|---|---|---|---|---|
| KIDMED | Model 1 | 0.07 | **0.006** | Body Mass | −0.24 | KIDMED = 8.90 − 0.03*Body Mass |
| MB Throw | Model 1 | 0.61 | **0.000** | MM | 0.78 | MB Throw = 1.09 + 0.18*MM |
|  | Model 2 | 0.72 | **0.000** | MM | 1.37 | MB Throw = 3.40 + 0.32*MM - 0.28*BMI |
|  |  |  |  | BMI | −0.67 |  |
| T-Half Test | Model 1 | 0.16 | **0.000** | Height | −0.40 | T-Half Test = 11.38 - 2.83*Height |
|  | Model 2 | 0.29 | **0.000** | Height | −0.38 | T-Half Test = 8.67 - 2.69*Height + 0.01* $\sum$3-skinfolds |
|  |  |  |  | $\sum$3-skinfolds | 0.37 |  |
|  | Model 3 | 0.33 | **0.000** | Height | −0.15 | T-Half Test = 7.06 - 1.10*Height + 0.02* $\sum$3-skinfolds - 0.03*MM |
|  |  |  |  | $\sum$3-skinfolds | 0.51 |  |
|  |  |  |  | MM | −0.32 |  |
| SJ | Model 1 | 0.21 | **0.000** | FM | −0.46 | SJ = 30.65 - 0.36*FM |
|  | Model 2 | 0.38 | **0.000** | FM | −0.42 | SJ = -17.83 - 0.33*FM + 28.16*Height |
|  |  |  |  | Height | 0.41 |  |
| CMJ | Model 1 | 0.29 | **0.000** | FM | −0.54 | CMJ = 36.74 - 0.47*FM |
|  | Model 2 | 0.51 | **0.000** | FM | −0.72 | CMJ = 24.42 - 0.64*FM + 0.53*MM |
|  |  |  |  | MM | 0.50 |  |
| Sprint | Model 1 | 0.42 | **0.000** | FM | 0.64 | Sprint = 3.88 + 0.03*FM |
|  | Model 2 | 0.57 | **0.000** | FM | 0.61 | Sprint = 6.59 + 0.03*FM - 1.58*Height |
|  |  |  | **0.000** | Height | −0.39 |  |
| YoYo distance | Model 1 | 0.29 | **0.000** | FM | −0.54 | YoYo distance = 1505 - 28.57*FM |
|  | Model 2 | 0.32 | **0.000** | FM | −0.52 | YoYo distance = 123 - 27.77*FM + 803*Height |
|  |  |  |  | Height | 0.17 |  |
| YoYo $VO_2$ max | Model 1 | 0.29 | **0.000** | FM | −0.54 | YoYo $VO_2$ max = 49.05 −0.24*FM |
|  | Model 2 | 0.32 | **0.000** | FM | −0.52 | YoYo $VO_2$ max = 37.43 −0.23*FM + 6.75*Height |
|  |  |  |  | Height | 0.17 |  |

**Notes.**

BMI, Body Mass Index; FM, Fat mass; $\sum$3-skinfolds, sum of 3-skinfolds; MM, Muscle mass; MB, Medicine ball; SJ, Squad Jump; CMJ, Counter Movement Jump. Values in bold are statistically significant ($p < 0.05$).

## Competing Interests

The authors declare there are no competing interests.

## Author Contributions

- David Romero-García conceived and designed the experiments, performed the experiments, analyzed the data, prepared figures and/or tables, authored or reviewed drafts of the article, and approved the final draft.
- Francisco Esparza-Ros conceived and designed the experiments, authored or reviewed drafts of the article, and approved the final draft.
- María Picó García performed the experiments, analyzed the data, prepared figures and/or tables, authored or reviewed drafts of the article, and approved the final draft.
- José Miguel Martínez-Sanz conceived and designed the experiments, performed the experiments, authored or reviewed drafts of the article, and approved the final draft.

- Raquel Vaquero-Cristóbal conceived and designed the experiments, performed the experiments, analyzed the data, authored or reviewed drafts of the article, and approved the final draft.

## Human Ethics

The following information was supplied relating to ethical approvals (i.e., approving body and any reference numbers):

The study was approved by the Ethics Committee of the University of Alicante (code: UA-2022-02-01).

## Data Availability

The raw data is available in the Supplementary File.

## Supplemental Information

Supplemental information for this article can be found online at http://dx.doi.org/10.7717/peerj.14329#supplemental-information.

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
