# Peer review of "Adherence to the Mediterranean diet, kinanthropometric characteristics and physical performance of young male handball players"

_PeerJ, doi:10.7717/peerj.14329_

## Round 0.1 · original submission · Major Revisions

Dear Authors, I suggest revising the manuscript according to the revivers' comments to improve your work. Your work is well-written and original and ensures knowledge about this argument.

·

Basic reporting

The work is well written and the aims of the study are well clarified by the authors. The English is clear and understandable to an international audience. The introduction lacks a robust and detailed description of the characteristics of the Mediterranean diet and why it should be better than other types of diet for the benefit of sports performance and body composition. Please clarify this point better from line 64 onwards.
The bibliography is well edited and up-to-date and the aims of the study are quite clear. The raw data is extensive and I thank you for providing it.

Experimental design

In general the methodology used is clear and well structured, with regard to the Yo-Yo test (line 143) is it possible to include an estimate of VO2max using the regression equation (Bangsbo et al. 2003) and then calculate any interactions and/or correlations with the other parameters? With regard to statistical analysis, the confidence interval for correlation coefficients must be included (Batterham AM; and Hopkins WG, 2000). The tables are well structured but the graphs representing the results obtained relative to the correlation coefficients are absent, can these be included?

Validity of the findings

The data are well represented and find confirmation in the international literature cited by the authors. Possible future research developments are well clarified in the discussion. Possible practical applications for practitioners in relation to the results obtained should be included in the conclusions.

Additional comments

In order of importance, it is necessary to clarify these points:

1. Better clarify the characteristics of the Mediterranean diet and clarify why it should be better than other diet plans;
2. Enter the confidence interval regarding the correlation coefficients;
3. Insert graphs representing the results obtained in relation to the correlation coefficients;
4. Clarifying the practical applications useful to practitioners derived from the results obtained;
5. Insert an estimate of VO2max in relation to the Yo-Yo test and insert it in the statistical analysis of the results obtained.

I congratulate the authors on their work, which is clearly original and well-structured, the result of good scientific planning with solid research hypotheses. Having clarified the points listed above, the article will be ready for publication in our journal.

Reviewer 2 ·

Basic reporting

I would like to thank you for submitting and give me the opportunity to review the manuscript entitled: “Adherence to the Mediterranean diet, kinanthropometric characteristics and physical performance of young male handball players”. The research topic undertaken by the authors is interesting in terms of expanding anthropometric knowledge and its relationship with other variables in the sport of handball. Nevertheless, some questions and concerns need to be answered and corrected before the formal acceptance of the manuscript.

A clear hypothesis is missing.

In the summary, add the sample size and specify the main anthropometric characteristics that show significant correlations with adherence to the Mediterranean diet.

In materials and methods, add the province(s) and country where the study was carried out. In line 101 add: “or legal guardians, failing this”, in line 120 specify the four basic measures.

In results, how do you explain that the infant category showed significantly lower values in all the variables analysed, included folds and perimeters, and significantly higher values in the percentage of fat mass and 3-fold sum than the rest of the categories?

In the tables, I advise you to carefully review the information at the foot of the tables, since in several tables the explanation of acronyms is missing and in some of them there are acronyms that do not appear in the tables. Revise. On the other hand, I also advise highlighting significant differences in bold, for a better visualisation of the results.

In the discussion with other studies, you make a lot of reference to the gender of the athletes, especially in the part on adherence to the Mediterranean diet. In this sense, and since your sample is only male, the comparison with these studies may not be very representative. What do you think about this? How could this weakness be solved?

And finally, and regarding the citations, I consider it redundant to duplicate the citation in the text (for example: line 254 appears Hermassi et al. (Hermassi et al., 2020)). If the journal rules allow it, I propose to change Hermassi et al. (Hermassi et al., 2020) to Hermassi et al., 2020.... Same proposal for subsequent citations of the same style.

Experimental design

No comment. All my comments are together in the basic reporting.

Validity of the findings

No comment. All my comments are together in the basic reporting.

Additional comments

No comment. All my comments are together in the basic reporting.

---

## Round 0.2 · accepted · Accept

Congratulations on the revised version.